# Cyanobacteria, Cyanotoxins, and Neurodegenerative Diseases: *Dangerous Liaisons*

**DOI:** 10.3390/ijms22168726

**Published:** 2021-08-13

**Authors:** Paola Sini, Thi Bang Chau Dang, Milena Fais, Manuela Galioto, Bachisio Mario Padedda, Antonella Lugliè, Ciro Iaccarino, Claudia Crosio

**Affiliations:** 1Department of Biomedical Sciences, University of Sassari, 07100 Sassari, Italy; sinipaoladrop@gmail.com (P.S.); bangchauykt@gmail.com (T.B.C.D.); faismilena@gmail.com (M.F.); galioto@uniss.it (M.G.); ciaccarino@uniss.it (C.I.); 2Department of Biochemistry, Hue University, Hue City 7474, Vietnam; 3Laboratory of Aquatic Ecology, Department of Architecture, Design and Urban Planning, University of Sassari, 07100 Sassari, Italy; bmpadedda@uniss.it (B.M.P.); luglie@uniss.it (A.L.)

**Keywords:** cyanobacteria, cyanotoxins, neurodegenerative diseases, ALS, Amyotrophic Lateral Sclerosis, PD, Parkinson’s Disease, AD, Alzheimer Disease, L-BMAA

## Abstract

The prevalence of neurodegenerative disease (ND) is increasing, partly owing to extensions in lifespan, with a larger percentage of members living to an older age, but the ND aetiology and pathogenesis are not fully understood, and effective treatments are still lacking. Neurodegenerative diseases such as Alzheimer’s disease, Parkinson’s disease, and amyotrophic lateral sclerosis are generally thought to progress as a consequence of genetic susceptibility and environmental influences. Up to now, several environmental triggers have been associated with NDs, and recent studies suggest that some cyanotoxins, produced by cyanobacteria and acting through a variety of molecular mechanisms, are highly neurotoxic, although their roles in neuropathy and particularly in NDs are still controversial. In this review, we summarize the most relevant and recent evidence that points at cyanotoxins as environmental triggers in NDs development.

## 1. Introduction

The aetiology and pathogenesis of neurodegenerative diseases (NDs) such as Alzheimer’s disease (AD), Parkinson’s disease (PD), and amyotrophic lateral sclerosis (ALS) are not fully understood. All these neurodegenerative disorders have a significant genetic contribution, although mendelian forms of NDs, attributed to rare gene mutations, may account only for up to 5–10% of the cases, and the remaining 90–95% are due to idiopathic mechanisms. Recent high-throughput genomic technologies have demonstrated that the NDs share common genetic factors, and microarrays and next-generation RNA-sequencing point to shared gene expression signatures, such as neuroinflammation genes [1], with further overlaps identified in genes related to RNA splicing and protein turnover between ALS and PD and mitochondrial dysfunction genes as a common theme between PD and AD. Moreover, a recent meta-analysis study on -omic data obtained at all gene expression levels reveals significant overlaps between the different diseases [2]. 

Patients affected by NDs share common genetic patterns, although a consistent percentage of sporadic cases may have causes other than or in addition to human hereditary factors. The non-genetic factors may include the involvement of a variety of environmental factors, such as toxins, produced naturally by microorganisms. Table 1 summarizes some of the most representative epidemiological data, verified by meta-analysis, linking NDs to environmental factors.

Cyanobacteria and microalgae synthesize significant quantities of toxins that can act via multiple molecular mechanisms [29,30]. Recent studies showing the presence of the neurotoxin β-N-methylamino-L-alanine (L-BMAA), produced by cyanobacteria and algal species, in the brain and cerebro-spinal fluid samples from patients with AD and ALS suggest that exposure to cyanotoxins may contribute to the development of human neurodegenerative diseases [27,31,32]. However, understanding the neurotoxic effects of L-BMAA and other microalgal neurotoxins and identification of pharmacological strategies to attenuate these harmful effects is needed.

Harmful algal blooms (HABs) represent a natural phenomenon caused by the growth of single or more species of phytoplankton at the same time. The harmful algal species (HAS) may belong to two different kingdoms of life, prokaryotic cyanobacteria and eukaryotic microalgae in waterbodies. In the last decades, HABs have had an evident increase in connection to human impacts such as eutrophication, aquaculture, hydrodynamic modifications in coastal systems, and global climate change [33]. Part of this observed HAB expansion reflects a better assessment of the current and past scale of the phenomenon, long obscured by scarce monitoring [34].

Over recent decades, it has been demonstrated that increasing anthropogenic activities, such as intensive agriculture and farming, industrialization, and urbanization, have led to the widespread eutrophication of inland and coastal ecosystems, resulting in a range of environmental, social, and economic issues due to the degradation of water resources [35,36]. Eutrophication causes shifts in the aquatic ecosystem’s state, leading to a loss of ecosystem goods and services [37]. In fact, the quantity and quality of nutrient inputs to a water body can have profound effects upon its ecosystem processes and structure, e.g., acting on its biogeochemistry and biodiversity and altering the water quality. Eutrophication has many negative effects, among which one of the most worrying is the increased growth of microalgae [38] and cyanobacteria [39,40] that interfere with the use of waters [41]. Their blooms contribute to a range of problems, including fish kills, foul odors, unpalatability of drinking water, and hazards for human health [40].

The nutrient supplies to water bodies originate from different sources, such as external inputs, including catchment drainage, groundwater, and the atmosphere, and internal inputs, such as release from sediments. Strong relationships have been demonstrated between total phosphorus inputs and phytoplankton production in freshwaters [42,43,44], where N_2_-fixing cyanobacteria often dominate, compensating for any deficit in nitrogen [45,46], as well for the intake of total nitrogen in estuarine [47] and marine waters [48,49], on a worldwide scale. Changes in nutrient supply ratios, particularly for mineral (N:P or N:Si) and organic forms (DOC:DON), are responsible for the rearrangement of phytoplankton assemblages in favor of dominant species, which can lead to the formation of blooms [50,51,52]. Despite progress in our knowledge of the mechanisms by which nutrients are supplied to ecosystems and the pathways by which different species absorb them, the connections between nutrient supply and bloom growth, as well as their potential toxicity or damage, remain poorly understood [53]. The increase in the abundance of algal prey is also responsible for the widespread heterotrophic and mixotrophic species among HAB [54,55]. The ecological success of a microalgae or cyanobacteria species is influenced by biological factors, such as the presence and abundance of other species, grazers [56], and abiotic factors, such as the flushing rate or water residence time, weather conditions, water mixing, and stratification. The overall impact of nutrient overabundance on hazardous algal species is strongly species-specific. Control and reductions of nutrients have been demonstrated as the only effective and structural solution to preventing phytoplankton biomass or HAB incidence [57].

The HAS, mainly represented by dinoflagellates, diatoms, and cyanobacteria, produce significant environmental impacts due to high biomass and/or toxin production (Figure 1).

Collectively, cyanotoxins and algal toxins have been implicated in an array of human diseases. In particular, the consumption of food contaminated by algal toxins results in various pathological conditions including seafood poisoning syndromes (diarrhetic shellfish poisoning—DSP, paralytic shellfish poisoning—PSP, neurotoxic shellfish poisoning—NSP, ciguatera fish poisoning—CFP, due to dinoflagellates; amnesic shellfish poisoning—ASP, due to diatoms). Human contact with aerosol or waterborne toxins can also have other minor deleterious impacts, such as dermatological or respiratory irritation [58]. Moreover, there is increasing epidemiological evidence of relationships between environmental toxins and neurodegenerative diseases, including ALS, AD, and PD [27,59,60].

Most of these pathological conditions are caused by neurotoxins, which show highly specific effects on the nervous system of animals, including humans, by interfering with nerve impulse transmission. Neurotoxins are a varied group of compounds, both chemically and pharmacologically. They vary in both chemical structure and mechanism of action and produce very distinct biological effects, which provide a potential application of these toxins in pharmacology and toxicology.

Whereas dinoflagellates and diatoms are found primarily in marine environments, cyanobacteria are usually considered the major HAS in freshwater ecosystems. Actually, their impacts on transitional aquatic ecosystems may increase due to global climatic change [61]. Cyanobacteria produce an impressive range of toxic secondary metabolites, the cyanotoxins, whose presence and concentration in the waters is both a relevant threat to human health and the environment and a substantial economic cost [62,63]. Cyanobacteria are ancient, cosmopolitan inhabitants of terrestrial environments and fresh, transitional, and marine ecosystems; they are photosynthetic and prokaryotic organisms, classified in 150 genera, over 40 of which include species that produce cyanotoxins [64]. Cyanobacteria are fundamental components of phytoplankton, and their competitiveness, which depends on both biological traits and environmental conditions, allows them to dominate the phytoplankton of eutrophic and hypereutrophic water bodies. Interestingly, cyanobacteria have been globally growing due to the increase of the geographical distribution, including in the Mediterranean region, frequency, and extent of their harmful blooms (cyano-HABs), which are expected to further increase due to climate change [65,66]. Exposure to cyanotoxins, responsible for acute or (sub)chronic poisonings of wild/domestic animals and humans, can follow multiple routes: i) orally, via drinking water or via consumption of health food tablets or other organisms that have accumulated the cyanotoxins along the food chains; ii) in labour or recreational water environments dermally; or iii) by inhalation exposure [67]. 

Cyanotoxins are grouped, according to the physiological systems, organs, tissues, or cells that are primarily affected, in neurotoxins, hepatotoxins, cytotoxins, irritants, and gastrointestinal toxins. Many cyanotoxins are also tumor promoters, with carcinogenic activity, and are the causative agents of serious health threats for humans [68].

The purpose of this review is to summarize the scientific information on the relationship between neurodegenerative disorders and cyanobacterial/dinoflagellates neurotoxins, classified according to [69], focusing on the experimental models used to test CTX toxicity.

## 2. Cyanobacterial and Dinoflagellates Neurotoxins 

According to [28,70], cyanobacterial and dinoflagellates neurotoxins can be divided in four main classes, based on their mode of action: 

saxitoxins (carbamate compounds, N-sulfocarbonyl compunds, decarbamyl compunds);

ciguatoxins;

anatoxins (anatoxin-a, homoanatoxin-a, guanitoxin);

β-N-methylamino-L-alanine (L-BMAA) and its isomers (2,4-diaminobutyric acid, 2,4-DAB and aminoethylglycine, AEG);

## 3. Saxitoxins and the Paralytic Shellfish Poisoning

Saxitoxin (STX) and its 57 analogues, collectively indicated as paralytic shellfish toxins (PSTs), are a family of molecules consisting of a tetrahydropurine group and two guanidinium moieties, produced by both cyanobacteria and dinoflagellates [70,71].

The most well-known and researched source of the PSTs is marine dinoflagellates (e.g., *Alexandrium*), which are filtered by invertebrates such as shellfish, crustaceans, and molluscs without being affected by the toxins. The toxins become concentrated in the invertebrates and are then ingested by human consumers, causing paralytic shellfish poisoning (PSP). There are strict safety guidelines for commercially produced seafood that establish a shellfish harvesting prohibition if toxin levels exceed a maximum of 800 μg STX eqv/1000 g edible tissue [72].

STX is one of the strongest natural neurotoxins, and it is also the most studied among PSTs [71]. STX is a reversible voltage-gated sodium channel blocker (Figure 2A) [73]. It crosses the blood–brain barrier and acts by blocking sodium channels in the central nervous system (CNS), therefore leading to paralytic effects [74]. A critical issue related to low-dose extended exposure of coastal communities who rely heavily on a seafood diet is the study of the molecular mechanisms underlying STX toxicity. Exposure to STXs of cultured primary murine motoneurons as well neuronal cell lines (PC12 and SH-SY5Y cell lines) induces a reduction in axonal growth that is dependent on the presence of voltage-gated sodium channel isoform Nav1.9 [66,75]. Interestingly, the pharmacological activation to increase the opening probability of NaV1.9 could be a way to stimulate axon regeneration and maintenance in human neurodegenerative pathology such as spinal muscular atrophy (SMA), in which a defect in synapse maintenance appears as a central pathophysiological mechanism.

A low dose of STXs induces an altered redox status that results in oxidative stress in different experimental paradigms, as reported in Table 2.

Recently, a proteomic study on murine neuroblastoma N2A cells identified different proteins altered upon low-dose saxitoxin exposure. The identified proteins are key regulators of cell apoptotic pathways, cell skeleton maintenance, membrane potentials, and mitochondrial functions [31]. Notably low doses of saxitoxins induce a decrease in voltage-dependent anion-selective channel 1 (VDAC1). VDAC1 is a multifunctional protein, expressed in the mitochondria and other cell compartments, that regulates the main metabolic and energetic functions of the cell (Ca2+ homeostasis, oxidative stress, and mitochondria-mediated apoptosis) [32]. Notably, VDAC1 represents the main mitochondrial docking site of many misfolded proteins, such as amyloid β and Tau in AD, α-synuclein in PD and several SOD1 mutants in ALS [33]. In AD post-mortem brains as well as in APP1 transgenic mouse models, VDAC1 was found to be over expressed in patients, and the possibility of decreasing it by using low doses of STX can be a fascinating therapeutic option [33].

## 4. Ciguatoxins

Ciguatoxins (CTXs) are polyether marine toxins known to activate voltage-gated sodium channels (NaV) and to cause one of the most widespread forms of nonbacterial food poisoning, named ciguatera. They are produced by dinoflagellates (i.e., Gambierdiscus) and reach humans via the food chain, with the consumption of fish that graze on reef macroalgae, including dinoflagellates that produce CTXs.

CTX-caused food poisoning was endemic only in tropical and subtropical areas, but it is spreading in Europe and Australia. Despite the high number of cases, estimated at around 50,000–500,000 cases per year, the prognosis is usually benign. In humans, more than 170 non-specific symptoms have been reported, although the most characteristic manifestations of ciguatera fish poisoning, found in all patients, are neurological symptoms, including paraesthesia and headache [81]. 

At present, more than 29 different CTX analogues have been identified, and they have been classified into three main groups that differ slightly in chemical structure according to the origin of the toxin: P for Pacific, C for the Caribbean, and I for Indian ciguatoxins [82] (Table 3).

Several lines of evidence suggest that chronic exposure to P-CTX-1 is associated with severe neurological manifestations in the peripheral nervous system (PNS) of some patients, suggesting that P-CTX-1 neurotoxicity, similar to other peripheral neuropathologies, primarily affects the PNS. In line with this hypothesis, it has been demonstrated, in mouse models, that the persistence of P-CTX-1 in peripheral nerves reduces the intrinsic growth capacity of peripheral neurons, resulting in delayed functional recovery after injury [83]. Moreover, P-CTX-1 has been shown to be a relatively non-selective activator of human NaVs subtypes (Figure 2A, displaying different functional effects on the different NaV subtypes, differentially expressed in peripheral sensory neurons [84]. It has been recently demonstrated that local application of 1 nM P-CTX-1 into the skin of human subjects induces a long-lasting, painful axon reflex flare and that CTXs are particularly effective in releasing calcitonin-gene related peptide (CGRP) from nerve terminals [85]. Significant alteration in CGRP expression has been also observed in the anterior horn of the spinal cord of familial ALS patients as well as in the transgenic mice expressing mutated human SOD1, one of the most-used ALS mice models [86]. In this ALS mouse model, the genetic deletion of CGRP accelerates muscle denervation and reduces cytotoxic neuroinflammation [87]. Interestingly, in the spinal cord of wobbler mice, a well-established model of motor neuron loss, an increase in mRNA of CGRP and its receptor, has been observed [88]. 

Additionally, CNS neuron physiology is altered upon CTXs exposure since synthetic ciguatoxin P-CTX-3C has been shown to have a profound effect on neuronal transmission in mice primary cortical neurons [89]. The transcriptomic analysis of cortical neurons exposed for different time points to P-CTX-3C led to the identification of different signaling pathways activated downstream to the activating NaVs [90].

P-CTX-3C induces cytotoxicity in SHSY5Y human neuronal cells, only in the presence of the Na^+^ channel activator (veratridine) and of the inhibitor of the Na^+^/K^+^ ATPase (ouabain), mimicking a realistic human in vivo situation [91].

Interesting results were obtained using a tetracyclic analogue of ciguatoxin-like toxin, gambierol, in cellular and animal models for AD. In fact, although gambierol exhibits a potent acute lethal toxicity in mice (minimal lethal dose: 50 μg/kg, ip), its tetracyclic truncated analogue in a mouse model for AD induces a decrease of amyloid β1−42 level, a reduction of tau phosphorylation, and a reduction in the N2A subunit of the N-methyl-D-aspartate (NMDA) receptor level [92].

## 5. Anatoxins

Anatoxins are water-soluble cyanotoxins (produced by different cyanobacterial genera, e.g., *Anabaena, Dolichospermum, Aphanizomenon*; Figure 3), lethal neurotoxins that can be classified into three main categories: anatoxin-a, its structural homologue homoanatoxin-a, and the unrelated guanitoxin, previously named anatoxin-a(s) [95]. 

Anatoxin-containing blooms have been found all over the world. As represented in Figure 2B, they have different physiological targets: (i) anatoxin-a is an alkaloid and an agonist of nicotine acetylcholine receptors (nAChRs), which are located both in the CNS as well as in the postsynaptic terminals of motor neurons, [96]; (ii) guanitoxin is an organophosphate that acts as an irreversible inhibitor of acetylcholinesterase (AChE, EC3.1.1.7) [28]. Notably, neuronal nAChRs are considered potential targets for the development of new therapeutic agents for the treatment of diverse disorders such as PD and AD [97,98], while AChE inhibitors have been demonstrated to be effective in slowing the clinical progression in AD patients [99].

Anatoxin-a was shown, at least in vitro, to induce inflammation and apoptosis in immune and brain cells [100], and it has been implicated in numerous animal poisonings worldwide. Up to date, there is no evidence of its toxic effects on the brain, and more detailed experiments are needed to find a link, if any, between anatoxin exposure and neurodegeneration.

## 6. Role of L-BMAA in Neurodegenerative Diseases

L-BMAA was isolated for the first time from the seeds of *Cycas circinalis* L. [54]. L-BMAA is a non-protein neurotoxic amino acid produced almost from all known groups of cyanobacteria including cyanobacterial symbionts (e.g., Nostoc) and free-living cyanobacteria (e.g., Anabaena, Microcystis; Figure 3), marine diatoms (e.g., Navicula, Skeletonema), and dinoflagellates (e.g., Gymnodinium) in the most various ecosystems worldwide [101].

Despite some contradictory opinions [102], an increasingly large body of experimental outcomes provides significant evidence that L-BMAA plays an important role in slow-developing neurodegenerative diseases, including ALS/Parkinsonism Dementia Complex (ALS/PDC) found on Guam islands, ALS, AD, and PD (review [32,59]).

ALS/PDC, specific to Guam and certain other Marianas islands of the Western Pacific, with symptoms of all three diseases, came to the attention of the scientific community during and after World War II. In the 1950s, for Chamorro residents of Guam and Rota, ALS, ALS-like conditions, and their death rates were estimated to be 50–100 times higher than in the United States and in other developed countries. From the late 1960s to the early 1980s, the incidence of both disorders had decreased. The main causes responsible for the decreasing incidence appeared to be ethnographic, social, and ecological changes, brought about by the rapid westernization of Guam. This change suggests that the cause of the ALS/PDC was not genetic but rather environmental [103]. 

Since the indigenous Chamorro people consumed cycad seed flour in food and in traditional medicine, Spencer et al. [104] first proposed the connection between the etiopathogenesis of ALS/PDC and the neurotoxin L-BMAA produced by the cyanobacteria of the genus *Nostoc*, which are symbiont of coralloid roots cycads. In a preliminary study, Spencer et al. showed that repeated oral administration of L-BMAA (0–81 mmol/kg daily) to *Macaca fascicularis* monkeys was able to induce a degenerative motor-system disease with features of ALS and parkinsonism. Pyramidal dysfunction, limb weakness, atrophy, upper-extremity tremors and wrist drop, bradykinesia, behavioral changes, and degeneration of lower motor neurons were observed [104]. 

A significant finding in the primate study of Spencer et al. was that, while early signs of motor-neuron dysfunction were observed in animal models fed with high doses of L-BMAA, extrapyramidal damage developed slowly with lower doses of L-BMAA. This led the authors to propose that chronic toxicity might be separate from acute toxicity [60]. Interestingly, L-BMAA is then biomagnified up the food chain from symbiotic cyanobacteria to cycads to *flying fox* of the genus *Pteropus mariannus*. Cox et al. [105] observed a 10,000-fold biomagnification of free L-BMAA and 50-fold biomagnification in total L-BMAA. These data suggested a mechanism that could produce sufficiently high doses of toxins to induce neurological disease in humans [106,107,108].

Biomagnification of L-BMAA may not be unique to Guam; indeed, Cox and colleagues [109] detected L-BMAA not only in the brain tissue of Chamorros who died from ALS-PDC but also in Alzheimer’s patients from Canada due to the capability of the neurotoxin to cross the blood–brain barrier through an active transport mechanism [110,111]. This finding suggests various ecological pathways for the bioaccumulation of L-BMAA in aquatic or terrestrial ecosystems.

L-BMAA is neurotoxic, and although different and multiple mechanisms of toxicity have been proposed (Figure 4), its involvement in neurotoxicity and neurodegeneration remains largely unidentified [112,113]. The neurotoxin is a non-lipophilic, non-essential amino acid that is present both in free and protein-bound forms. Weiss and Choi discovered that L-BMAA had activity in vitro only when a physiological concentration (10 mM and higher) of bicarbonate ions (HCO3^-^) was co-present in the cell culture media. L-BMAA’s carbamate adduct, named β-carbamate, presents structural similarities to glutamate that may lead to neuronal degeneration via a mechanism regulated by the activation of excitatory amino acid (EAA) receptors and/or glutamate transporters through a three-fold mechanism [114,115]. At a glutamatergic synapse, β-carbamate binds to ionotropic (NMDA and AMPA/kainate receptors) receptors (iGluR) and metabotropic receptors (mGluR). Their activation induces a significant increase in intracellular Ca^2+^, directly via iGluR and indirectly via mGluR (via phospholipase C signaling) [116], promoting mitochondrial reactive oxygen species (ROS) generation and endoplasmic reticulum (ER) stress [113]. This excitotoxicity of postsynaptic neurons typically leads to neuronal death. Besides being part of the pathogenic mechanism leading to ALS, excitotoxicity could be responsible for the selective vulnerability of motoneurons during the progression of the disease [117,118].

Liu et al. in 2009 found that L-BMAA inhibits the cystine/glutamate antiporter (system Xc−-mediated cystine uptake, which leads to glutathione depletion and increased oxidative stress. In a cyclical system, L-BMAA seems to drive the release of glutamate through the Xc-system, which induces toxicity through the activation of the mGluR5 receptor. This transport may be the cause of L-BMAA accumulation in cells [118]. 

Once in the cytoplasm, the neurotoxin may probably be misincorporated in place of serine or alanine in neosynthesized cellular proteins. L-BMAA might also be associated with proteins through non-covalent bonds. The insertion of L-BMAA and other non-protein amino acids into proteins may generate protein disfunction, misfolding, and/or aggregation. Although further research is required concerning L-BMAA incorporation into proteins, L-BMAA is incorporated into proteins in place of L-serine [119], and a large portion of L-BMAA is protein-bound (60- to 130-fold greater amount) compared to L-BMAA detected in the free [119,120].

This incorporated L-BMAA in brain tissues may function as an endogenous neurotoxic reservoir that can slowly release free L-BMAA, causing neurological damage over years or even decades, which may explicate the observed long-latency period for neurological disease onset among the Chamorro people [32]. 

Protein misfolding often leads to the formation of insoluble aggregates, and anomalous accumulation of aggregates in the affected tissues is one of the main pathological changes observed in neurodegenerative diseases. In ALS, this phenomenon involves biological markers including TDP-43 (TAR DNA-binding protein 43), a protein encoded by the TARDBP gene, located in the cell nucleus of most tissues. In physiological conditions, TDP-43 shuttles between the nucleus and cytoplasm, and it is involved in various steps of RNA biogenesis and processing such as alternative splicing [121,122]. In pathological conditions, TDP-43 is hyperphosphorylated, ubiquitinated, and cleaved to generate C-terminal fragments, and it was identified as the main component of ubiquitinated inclusions in post-mortem tissues of ALS patients and patients with frontotemporal dementia [122,123].

Triggers with L-BMAA result in TDP-43 overexpression and aggregation in several in vitro and in vivo models: SH-SY5Y cell lines [124] and primary neurons (rats [124,125], mice [126,127], and zebrafish [128]). These specific forms of TDP-43 are present in patients with neurodegenerative diseases such as ALS and FTD.

The protein misincorporation of L-BMAA could affect protein-folding and successive accumulation of misfolded proteins into lysosomes [119]. This anomalous protein-synthesis is also supposed to lead to cell stress at the endoplasmic reticulum, independent of L-BMAA high concentration effects such as excitotoxicity and oxidative stress, deregulation of the reduction/oxidation systems, and an activation of some pro-apoptotic caspases like caspase-12 [129]. The resulting dysregulated protein homeostasis with low non-excitotoxic concentrations could be a contributing factor in the scenario of chronic L-BMAA exposure that may lead to late onset and slow progression of neurodegenerative diseases [129]. 

Moreover, L-BMAA leads to the activation of transcription factors known to be involved in the regulation of oxidative stress and cellular senescence such as X-box binding protein 1 and nuclear factor 2 erythroid like 2 [130]. Interestingly, the same high levels of these transcriptional regulators have been detected in the brains of patients with ALS, PD, AD, and front temporal dementia [131,132]. 

Numerous investigators used in vitro approaches to assess the potential role of L-BMAA on mammalian CNS models. It should be noted that most in vitro investigations needed high L-BMAA concentrations (≥100 μM) to produce cellular damage and toxicity (Table 4). These concentrations are not physiologically appropriate, and consequently the results are extremely difficult to interpret compared to in vivo responses. Therefore, numerous studies identified a possible mechanism of toxicity at a cellular level but are incomplete in relating the effects to L-BMAA environmental exposures.

Chiu and colleagues [114] reported NMDA receptor-mediated increases in intracellular calcium ions, ROS production, DNA damage, and neuronal death in primary human neuronal cells prepared from foetuses following exposure to L-BMAA, with the lowest toxic concentration in the presence of bicarbonate reported to be 400 µM [133]. The neuron-like cell lines are frequently chosen for their characteristics. SH-SY5Y, from human metastatic neuroblastoma, has dopaminergic, cholinergic, glutamatergic, and adenosinergic features; clonal rat pheochromocytoma cell line PC12, differentiated with nerve growth factor, is a recurrent model to study receptor-mediated excitotoxicity [134]. In order to investigate independent excitotoxic mechanisms, non-neuronal cells have also been used, but immortalized cells are significantly different from physiological characteristics in neurons; thus, numerous studies were made on primary neuronal cultures (Table 4). 

The production of L-BMAA is not limited to cycad seeds, and the risk to exposure to this neurotoxin is not confined to Guam. In some locations, cyanobacteria are directly consumed by people. In the mountains of Peru, Cyanobacteria *Nostoc commune* Vaucher ex Bornet and Flahault (with a L-BMAA concentration of 10 μg/g) are collected in the highland lakes by the indigenous people, who call them llullucha [149]. Indigenous people eat them directly, sell them in markets, and add them to salads, soups, or meat dishes. Direct dietary intake is not the only possible mode of exposure to cyanobacterial neurotoxins. Inhalation as a systemic delivery route has been demonstrated for microcystins in nasal swabs and blood samples from people at risk of swallowing water or inhaling spray while swimming, water skiing, jet skiing, or boating during algal blooms [150]. In 2009, a causative link was hypothesized between the inhalation of L-BMAA, present in soil crusts dominated by cyanobacteria and detected in desert dust, and the higher incidence of ALS observed in the Gulf war veterans younger than 45 years old [151], but experiments in rat models observed significant biochemical responses to L-BMAA only at extremely high (non-physiological) concentrations [152].

Notably, L-BMAA misincorporation into neuroproteins produces protein misfolding and is inhibited by L-serine [108,139] that was proposed as a potential therapeutic option for ALS ([153] phase 2 ClinicalTrials.gov Identifier: NCT03580616), AD (Phase 2 ClinicalTrials.gov identifier: NCT03062449) and hereditary sensory autonomic neuropathy type I (HSAN1) [154]. The molecular mechanism underlying L-serine neuroprotection is not fully elucidated and can be independent of L-BMAA-mediated neurotoxicity [139]. 

## 7. Conclusion 

An increasing cyanobacteria abundance is expected due to climate change and eutrophication, worsening the cyanotoxins issue and urging quick prevention and mitigation actions. Cyanobacteria detection in natural water samples with cyanotoxins (CTXs)-level determination should become a priority to prevent uncontrolled human exposure. Although pathophysiological mechanisms underlying ND is far from being completely understood, the link between CTX exposure and neurodegeneration is now widely accepted by the scientific community. Apart from the well-described via of CTXs exposure (ingestion, dermal contact, biomagnification), it could be critical also to evaluate the presence of cyanobacteria in gut microbiota. In this respect, in the last 10 years, a growing recognition within the scientific and medical communities points at the “microbiota–gut–brain axis” as a key element in neurodegenerative process (Table 1 and [155] for a comprehensive review). Different lines of investigation have suggested that some species of cyanobacteria are present in small numbers in the gastrointestinal tract, and through the production of specific CTX, they could be considered potentially responsible for inducing neurodegeneration [155,156,157]. At present, only in PD patients, a specific decrease in cyanobacteria (Family Aphanizomenonaceae, Genus Dolichospermum) has been reported [158]. The lack of data for other neurodegenerative disorders can be linked to the low abundance of Cyanobacteria in biological samples tested. Nevertheless, human beings can be, via dietary sources, chronically exposed to cyanotoxins and/or other algal toxins, single or in combinations, which can alter different cellular processes and activate specific immune responses, chronic mild gut inflammation, and ultimately neurodegenerative processes [59].

The present review aims to emphasize the relationship between the increasing number of HABs and eutrophication with the molecular evidence linking CTXs to neurodegeneration. A multidisciplinary approach is required to mitigate the human health risks and to fill different scientific gaps. It is of particular interest to test the hypotheses whether CTXs in water samples are linked to their trophic state, to cyanobacteria abundance and/or species composition living there, and finally at the molecular level, to definitively establish the contribution of CTX chronic exposure to neurodegeneration.

## Figures and Tables

**Figure 1 ijms-22-08726-f001:**
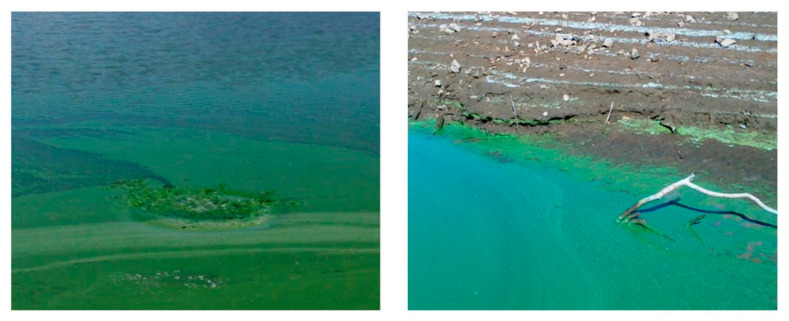
Evidence of intense cyano-HABs in Mediterranean artificial lakes (Sardinia; Lake Bidighinzu, on the left; Lake Posada, on the right). Cyanobacterial cell accumulation along shorelines, especially due to winds action, provokes blue-green colored waters.

**Figure 2 ijms-22-08726-f002:**
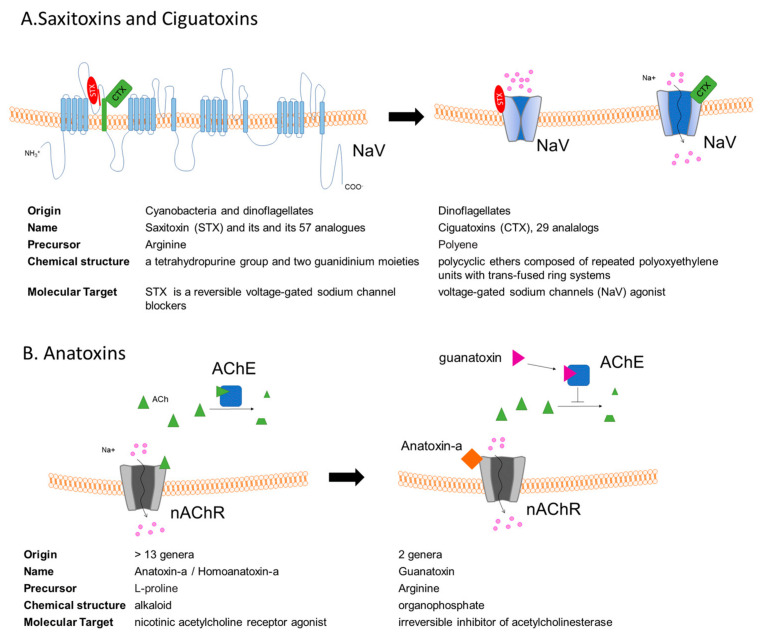
(**A**)Voltage-gated sodium channel (NaV) is the target of both saxitoxins (STXs) and ciguatoxins (CTXs). STX binding induces a block in Na^+^ conduction, while CTX binding slows down NaV inactivation. (**B**) Anatoxins act on the nicotinic acetylcholine receptor (nAchR): anatoxin-a is a nAChR receptor agonist, mimicking the binding of its natural ligand, acetylcholine (Ach), while guanatoxin inhibits acetylcoline esterase (AChE), inducing ACh accumulation at the neuromuscular junction.

**Figure 3 ijms-22-08726-f003:**
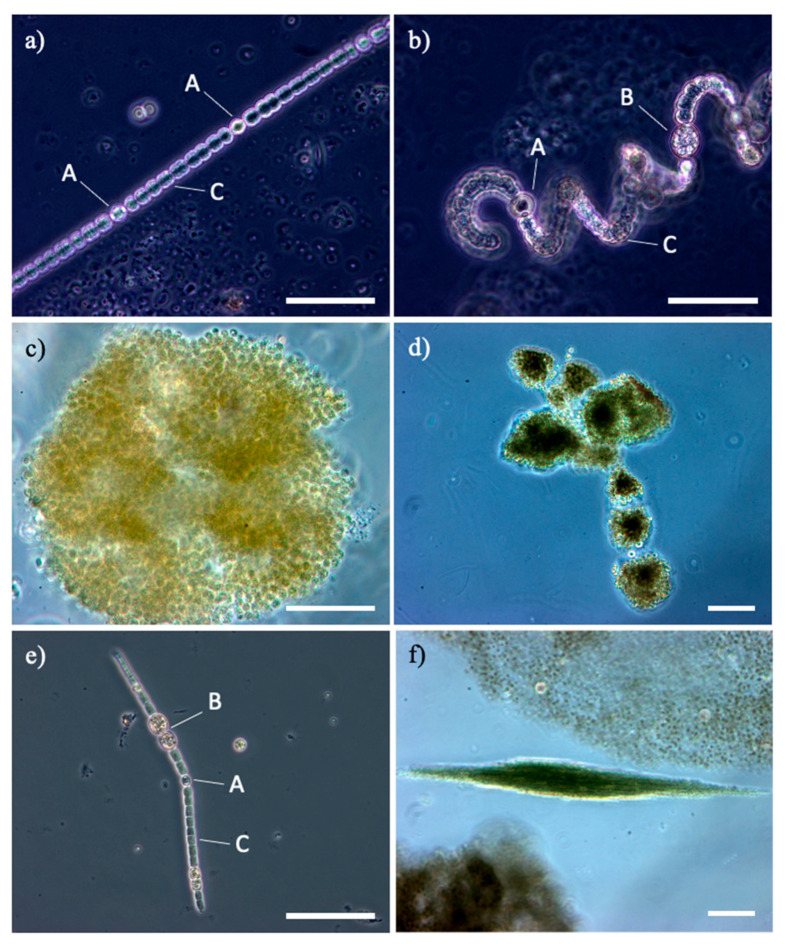
Cyanobacteria genera, potentially toxins producers in Mediterranean artificial lakes: (**a**,**b**): different species of Dolichospermum from Lake Bidighinzu; (**c**,**d**): different species of Microcystis from Lake Liscia and Lake Monte Lerno; (**e**,**f**): different species of Aphanizomenon, in single tricome from Lake Temo and in fascicle of tricomes from Lake Liscia. A: heterocyst, B: akinete; C: vegetative cells; bar 50 µm.

**Figure 4 ijms-22-08726-f004:**
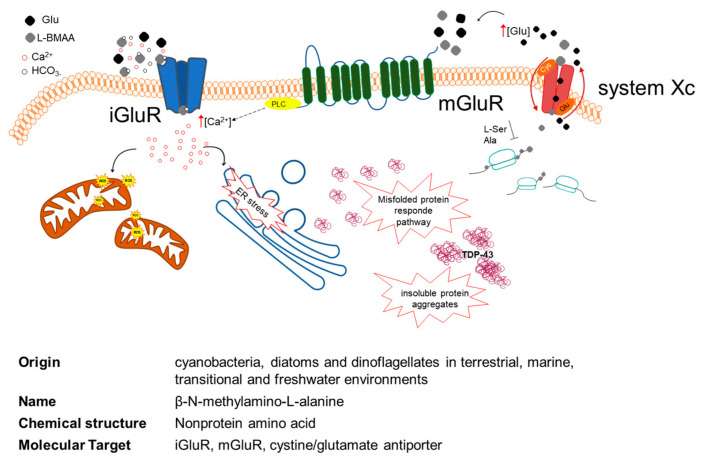
Multiple mechanisms of L-BMAA cellular toxicity. L-BMAA in the presence of bicarbonate ions (HCO3-) forms L-BMAA’s carbamate adduct, named β-carbamate, and binds to ionotropic (iGluR) and metabotropic (mGluR) receptors. The activation of iGluR and mGluR leads to a significant increase in intracellular Ca_2_^+^, directly via iGluR and indirectly via mGluR (PLC signaling). This Ca_2_^+^ increase promotes mitochondrial reactive oxygen species (ROS) generation and endoplasmic reticulum (ER) stress. L-BMAA inhibits the cystine/glutamate antiporter (system Xc-)-mediated cystine uptake, which leads to glutathione depletion and increased oxidative stress. Once in the cytoplasm, the toxin is likely to be inserted into the neosynthesized cellular proteins and to prompt protein misfolding that often leads to the formation of insoluble aggregates, containing among other proteins TDP-43. iGluR: ionotropic glu receptors; mGluR: metabotropic glu receptors; PLC: phospholipase C; TDP-43: TAR DNA-binding protein 43.

**Table 1 ijms-22-08726-t001:** Environmental factors in neurodegenerative diseases.

Environmental Factors	Effects	Diseases	Reference
**Heavy metals**	Lead (crosses the blood–brain barrier and accumulates in neuronal and glial cells)	ALS ^1^	[3,4]
	Aluminium	AD ^2^	[5,6]
	Manganese	PD ^3^	[6,7]
**Pesticide**	Pentachlorobenzene	ALS	[8]
	Rotenone and paraquat	ALS, PD	[9,10]
	Organophosphate pesticides	ALS, PD, AD	[11]
**Electromagnetic fields**	Contradictory results	ALS, AD	[12,13,14]
**Smoking**	Protective	PD	[15,16]
	Risk factor	AD, ALS	[17,18]
**Physical activity**	Protective	PD	[19]
**Body mass index and nutritional state**	Lower nutritional parameters	AD	[20]
**Microbiota structure** **and dysfunction of the gut–brain axis**	*Akkermansia muciniphila* reduces symptoms; *Ruminococcus torques* and *Parabacteroides distasonis*	ALS	[21,22]
	Suppression of *Prevotellaceae* and anti-inflammatory genera; blooming of pro-inflammatory *Proteobacteria*, *Enterococcaceae,* and *Enterobacteriaceae*	PD	[23,24,25]
	Suppression of anti-inflammatory taxa such as *Eubacterium rectale* and a profusion of pro-inflammatory taxa such as *Escherichia* and *Shigella*	AD	[24,26]
**Cyanobacteria and cyanotoxins**	Risk factors	ALS, PD, AD	[27,28]

^1^ ALS, Amyotrophic Lateral Sclerosis, ^2^ AD, Alzheimer’s Disease, ^3^ PD, Parkinson’s Disease.

**Table 2 ijms-22-08726-t002:** Saxitoxins treatment in different experimental models.

Experimental Model	Saxitoxins Exposure Protocol	Effects	Reference
primary neuron culture from tropical freshwater fish	0.3–3.0 mg L^−1^ 24h	oxidative stress, neurotoxicity, genotoxicity and apoptosis	[76]
murine neuroblastoma N2A	0–256 nM 24–48 h	high levels of ROS generation mild cytotoxic or apoptotic effects	[77]
rainbow trout fish cell line RTG-2	0–256 nM 24–48 h	mild cytotoxic or apoptotic effects	[77]
human primary astrocytes		high levels of ROS generation reduced cell survival	[78]
zebrafish embryos	0.05–0.1 µM	adverse effect on development of zebrafish embryos, oxidative stress-induced apoptosis	[79]
mouse neonate brain	single intraperitoneal 7.5 μg kg^−1^ body weight in pregnant mice	increased proliferation of OPCs, but not maturation process of these cells	[80]

**Table 3 ijms-22-08726-t003:** Ciguatoxins treatment in different experimental models.

Experimental Model	Ciguatoxins Exposure Protocol	Molecular Target	effects	Reference
SH-SY5Y	25 pM–100 nM P-CTX-3C short-(4–24 h) and long-term exposure (10 days)		cytotoxic effect, alterations of the mitochondrial metabolism, cell morphology, and [Ca2^+^]i	[91]
primary cortical neurons	5 nM CTX3C 6-24- 72 h		gene expression alteration mediated by voltage-gated sodium channel	[90]
C57BL/6 mice	shallow intraplantar (i.pl.) injection of P-CTX-1 (1–10 nM)	Nav 1.8 and TTXs Nav subtypes are effectors of ciguatoxin-induced cold allodynia	spontaneous pain	[93]
transgenic mice and rat	0.01–31 nM P-CTX-1 (>95% purity) isolated from moray eel (Gymnothorax javanicus) liver	NaV1.9	release of calcitonin-gene related peptide (CGRP) from nerve terminals	[85]
C57BL/6 mice	(0.26 ng/g body weight) intraperitoneally on day 0 followed by second exposure on day 3 P-CTX-1 (isolated and purified from moray eels)		irreversible motor deficit in 4-month pre-exposed mice following peripheral nerve injury astrogliosis and excitotoxic neuronal cell death via the activation of caspase 3 in motor cortex	[94]

**Table 4 ijms-22-08726-t004:** L-BMAA treatment in different experimental models.

Experimental Model	L-BMAA Exposure Protocol	Molecular Target	Reference
SH-SY5Y	3 mM plus antagonist for kainate/AMPA receptors 5 days	low neurotoxicity of BMAA and weak action at glutamatergic receptors	[135]
	0.1 mM 48h	Low non-excitotoxic BMAA concentrations induce effects on the ubiquitin/proteasome system not ROS-related	[129]
	3–10 mM 48h	decrease cell viability in a dose-response manner and evoke alterations in GSK3β and TDP-43	[136]
	0.5 mM 24h–48h–72h	Increased caspase-3 activity and cathepsins, ER stress	[137]
	0.05–0.25–1 mM 24 h	alterations in alanine, aspartate, and glutamate metabolism	[138]
	0.1–1 mM 24–48 h	autophagy	[139]
	3 mM 48h	disrupts mitochondrial metabolism	[140]
PC12	2 mM 6–12 h	apoptosis and mGluR1 increase	[141]
	0.4–1 mM 48h	promoted cell death and axon-like outgrowth	[142]
NSC-34	0.1–1 mM 72 h	exposure to BMAA causes protein misfolding, ER stress, induction of the UPR, disruption of the mitochondrial function	[130,141]
NIH/3T3	1–3 mM 48–96 h	L-BMAA causes arrest of cell cycle progression at the G1/S. No evidence of cell membrane damage, apoptosis, or ROS overproduction	[143]
primary cortical neurons	3 mM 1 h 20 mM HCO3^-^	L-BMAA activity is dependent on HCO3^-^, resulting in a destruction of cortical neuronal population.	[115] [144]
primary cerebellar granule cells colture, rat	up to 3 mM 24–48h	L-BMAA induced both necrotic- and apoptotic-like cell death	[145]
primary neurons and astrocytes cortical cell cultures, fetal mouse	3–10 mM 3–24h 0.1 mM 48h	enhancement death of cortical neurons damaged by other insults; oxidative stress, Wallerian-Like Degeneration	[146,147,148]
neural stem cells	50 µM–3 mM 24 h	apoptosis, cellular differentiation, neurite outgrowth, and DNA methylation	[133]

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
