# Peer review of "Cyanobacteria, Cyanotoxins, and Neurodegenerative Diseases: Dangerous Liaisons"

_ijms, 2021, doi:10.3390/ijms22168726_

Round 1
Reviewer 1 Report
This paper provides a relatively complete review of the literature in this important area of the etiology of the various neurodegenerative diseases. They do not report much in the field of Parkinson disease, largely because there is not much known in the way of gene-environmental interaction, apart from the protective effect of smoking. They should survey the literature to provide a fuller coverage of Parkinson disease.
The authors have included many of the papers from Dr. Paul Cox's group, but have not included discussion of the mammalian (vervet) model (Cox et al Proc Roy Soc B 2016, 83. and Davis et al J Neuropath and Exp Neurol 2020, 79, 393). These papers are the most relevant to the potential role of BMAA in human neurodegenerations, particularly in regard to treatment with L-serine.
The authors concentrate on climate change as being the most important cause of HABs, but should also review the importance of eutrophication.
The authors say on line 263 that BMAA is extremely neurotoxic, which is belied by their comment in line 325. In fact, there are many other compounds that are more neurotoxic than BMAA, certainly acutely.
Author Response
First of all, we would like to thank the reviewers for their suggestions, which we have used to improve the quality of our paper. As suggested by the Editors we added four figures and we implemented the references. Please find enclosed a point-by-point response to the suggested revisions.
Referee 1
This paper provides a relatively complete review of the literature in this important area of the aetiology of the various neurodegenerative diseases. They do not report much in the field of Parkinson disease, largely because there is not much known in the way of gene-environmental interaction, apart from the protective effect of smoking. They should survey the literature to provide a fuller coverage of Parkinson disease.
We thank the referee for the suggestion: we added a new table (Table 1) summarizing the environmental risk factor for NDs, including Parkinson Disease.
The authors have included many of the papers from Dr. Paul Cox's group, but have not included discussion of the mammalian (vervet) model (Cox et al Proc Roy Soc B 2016, 83. and Davis et al J Neuropath and Exp Neurol 2020, 79, 393). These papers are the most relevant to the potential role of BMAA in human neurodegenerations, particularly in regard to treatment with L-serine
As suggested by the referee we added the indicated papers from Dr. Paul Cox's group.
The authors concentrate on climate change as being the most important cause of HABs, but should also review the importance of eutrophication.
We thank the referee for the suggestion: we expand the Introduction, lighting up the attention on the importance of eutrophication.
The authors say on line 263 that BMAA is extremely neurotoxic, which is belied by their comment in line 325. In fact, there are many other compounds that are more neurotoxic than BMAA, certainly acutely.
We apologize for our misleading use of “extremely neurotoxic”, referred to L-BMAA
Reviewer 2 Report
The authors have written a nice mini-review on cyanobacteria and cyanotoxins. Prior to publication, the following issues should be addressed.
- Abstract, line 16: "as consequence" should be "as a consequence"
- Abstract, line 19: "neurotoxic although" should be "neurotoxic, although"
- Introduction: line 27: "Alzheimers" should be "Alzheimer's"
- Introduction, lines 50-51: "a single or more" should be "individual or multiple"
- Introduction line 78: "changes" should be "change"
- Introduction, line 86: "hypertrophic"- do the authors mean "hypereutrophic"?
- Introduction, line 96: "cells that primarily affected" should be "cells that are primarily affected"
- line 108: "cyguatoxins" should be "ciguatoxins"
- line 109: anatoxin-a(S). This has now been named "guanitoxin" by Fiore et al and this manuscript should be updated.
- Line 113: There are more than 50 variants of saxitoxins and the authors should find and reference the appropriate number.
- Line 117 and elsewhere: Genera are not italicized. Please check the journal to determine whether genera and species should be in italics.
- Line 123: This sentence is confusing. Are the authors saying that saxitoxin is one of the most toxic compounds of the saxitoxin/PST class? Please clarify.
- Line 125: "and by blocking" should be "and act by blocking"
- Line 126: "therefore lead" should be "leading"
- Line 130: "depending on the presence of voltage" should be "dependent on the presence of the voltage"
- Line 148: "overexpressed patients" should be "over expressed in patients"
- Line 151: "Cyguatoxins" should be "Ciguatoxins"
- Lines 174 and 201: "mice" should be "mouse"
- Line 183: delete "in" before "SOD1"
- Lines 195-197. This sentence is confusing- please clarify
- Line 207: "cyanobacteria" should be "cyanobacterial"
- Lines 209-213. Please rename anatoxin-a(S) as guanitoxins. This is also important as the authors state that the anatoxins are targets of acetylcholine receptors. This is true for anatoxin-a, but guanitoxin (anatoxin-a(S)) is an acetylcholine esterase inhibitor. Please clarify the paragraph.
- Line 214: should "in vitro" be in italics?
- Line 216-217: "experiment" should be "experiments"
- Lines 219-220: BMAA was originally isolated from Cycas circinalis by Vega and Bell (1967). The source of these seeds is not mentioned. Remove reference to India and Sri Lanka.
- Line 225: The paper by Chernoff et al. is a review and not primary data. Therefore, please change the word "data" to "opinion"
- Line 230: Change "of scientific" to "of the scientific"
- Line 234: Change "disorders had decreased" to "disorders decreased"
- Line 241: change "symbiont of coralloid roots cycads" to "symbionts of cycad coralloid roots"
- Line 247: "Spencer" should be "Spencer et al."
- Line 258: change "capability of neurotoxin" to "capability of the neurotoxin"
- Line 259: The sentence concerning active transport needs referencing.
- Line 270: "through three-fold" should be "through a three-fold"
- Line 289: Dunlop et al. showed that BMAA is misplaced for L-serine in proteins. Therefore the mechanism is not elusive- it would be better to replace with "Although further research is required concerning L-BMAA incorporation into proteins, L-BMAA is incorporated into proteins in place of L-serine (Dunlop et al., 2013) and a large portion of L-BMAA is protein bound (60 to 130 fold greater amount) compared to L-BMAA detected in the free....."
- Line 302: "processing such alternative" should be "processing such as alternative"
- LIne 307 and elsewhere through the manuscript: italicize in vitro and in vivo?
- Line 339: "mechanism" should be "mechanisms"
- Line 343: "to neurotoxin" should be "to this neurotoxin"
- Line 347: "Indigenous eat" should be "Indigenous people eat"
- Line 349: "exposition" should be "exposure"
- Line 350: "The inhalation as systemic delivery route it has" should be "Inhalation as a systemic delivery route has"
- Line 351: "risk for swallowing" should be "risk of swallowing"
- Line 352-3: "In 2009 it was hypothesized causative link between" should be "In 2009 a causative link was hypothesized"
- Line 367: Are the authors using CTX as cyanotoxin or ciguatoxin? Please clarify
- Lines 370-375. The sentence are a little confusing. Please check and clarify.
Author Response
First of all, we would like to thank the reviewers for their suggestions, which we have used to improve the quality of our paper. As suggested by the Editors we added four figures and we implemented the references. Please find enclosed a point-by-point response to the suggested revisions.
Referee 2
The authors have written a nice mini-review on cyanobacteria and cyanotoxins. Prior to publication, the following issues should be addressed.
We thank the referee for the deep revision of our review. As highlighted in the text we revised the manuscript according to referee’s suggestions and we clarify the misleading sentence (Line 123, 195-197; 209-213; 367; 370-375)
Line 113: There are more than 50 variants of saxitoxins and the authors should find and reference the appropriate number.
We apologize for the mistake. Two references were added
Line 117 and elsewhere: Genera are not italicized. Please check the journal to determine whether genera and species should be in italics.
Genera are now italicized according to IJMS editorial policy
Line 214: should "in vitro" be in italics?
According to IJMS editorial policy, “in vitro” can be written without using italics